# Catalytic Reductive Amination of Aromatic Aldehydes on Co-Containing Composites

**Vladyslav V. Subotin** [1,2], **Vitalii M. Asaula** [2], **Yulian L. Lishchenko** [1,2], **Mykyta O. Ivanytsya** [1,2], **Olena O. Pariiska** [2], **Sergey V. Ryabukhin** [1,3,4], **Dmitriy M. Volochnyuk** [1,3,4] and **Sergey V. Kolotilov** [2,3,*]

1   Enamine Ltd., Chervonotkatska Street 78, 02094 Kyiv, Ukraine
2   L.V. Pisarzhevskii Institute of Physical Chemistry of the National Academy of Sciences of Ukraine, Prosp. 7 Nauky 31, 03028 Kyiv, Ukraine
3   Institute of High Technologies, Taras Shevchenko National University of Kyiv, Volodymyrska Street 60, 01601 Kyiv, Ukraine
4   Institute of Organic Chemistry, National Academy of Sciences of Ukraine, Murmanska Street 5, 02660 Kyiv, Ukraine
*   Correspondence: s.v.kolotilov@gmail.com

**Abstract:** The performance of a series of cobalt-based composites in catalytic amination of aromatic aldehydes by amines in the presence of hydrogen as well as hydrogenation of quinoline was studied. The composites were prepared by pyrolysis of $Co^{II}$ acetate, organic precursor (imidazole, 1,10-phenantroline, 1,2-diaminobenzene or melamine) deposited on aerosil ($SiO_2$). These composites contained nanoparticles of metallic Co together with *N*-doped carboneous particles. Quantitative yields of the target amine in a reaction of *p*-methoxybenzaldehyde with *n*-butylamine were obtained at $p(H_2)$ = 150 bar, T = 150 °C for all composites. It was found that amination of *p*-methoxybenzaldehyde with *n*-butylamine and benzylamine at $p(H_2)$ = 100 bar, T = 100 °C led to the formation of the corresponding amines with the yields of 72–96%. In the case of diisopropylamine, amination did not occur, and *p*-methoxybenzyl alcohol was the sole or the major reaction product. Reaction of *p*-chlorobenzaldehyde with *n*-butylamine on the Co-containing composites at $p(H_2)$ = 100 bar, T = 100 °C resulted in the formation of N-butyl-N-*p*-chlorobenzylamine in 60–89% yields. Among the considered materials, the composite prepared by decomposition of $Co^{II}$ complex with 1,2-diaminobenzene on aerosil showed the highest yields of the target products and the best selectivity in all studied reactions.

**Keywords:** cobalt; composite; catalysis; amination; hydrogen; aldehyde; quinoline; activity

## 1. Introduction

Reductive amination is an important reaction widely applied for the transformation of the carbonyl compounds to different types of amines—important building blocks for medicinal chemistry [1,2]. It was estimated that such a method was used in ca. 25% cases of C–N bond formation cases for the pharmaceutical industry [3]. Reductive amination is based on the reaction of aldehydes or ketones with amines (or ammonia) in the presence of a reducing agent, such as borohydride or similar compounds with a B-H bond [4–7], silanes [8–11], formic acid [12], isopropanol [13], or hydrogen [14–17], and in many cases requires the additional use of the catalyst. It is accepted that in the first step of the reductive amination, the carbonyl compound reacts with the amine forming hemianinal, which can be hydrogenated directly to the target amine or can eliminate water giving azomethine (Schiff base) or enamine, depending on the amine, which undergoes further hydrogenation [18–20]. Use of hydrogen as a reducing agent for the amination appears to be the most expedient due to the absence of toxic reagents and a minimal quantity of waste [15,21].

Amination of the carbonyl compound with the use of hydrogen is normally performed in the presence of the catalyst, usually the one based on Pd [16,17,21]. In view of the growing price of Pd and its toxicity (which implies very strict demands for purification of

the products from traces of the catalyst [22]), the search for new platinum-group-metals-free catalytic systems is an important task. Several alternatives to the catalysts, contain platinum group metals, were considered for the amination. Historically, nickel appears to be the first catalyst used for such a purpose [14], and nickel-containing systems were considered as promising catalysts for the amination of aldehydes in recent studies [21,23–28]. Several Co-based [29–32] and Fe-based [33] catalysts were also proposed recently for the same purpose.

It can be expected that all catalysts of hydrogenation by $H_2$ (selected recent examples can be found in [34–36]) or transfer hydrogenation catalysts [37–40] have potential for use in the catalytic reductive amination processes. However, despite large quantities of the reported metal-containing hydrogenation catalysts, only a limited number of such systems was tested as the catalysts for the amination processes. Information about the catalytic activity of the composites, containing 3d metals, in amination processes is scarce compared to hydrogenation reactions. The search for cheap and readily available catalysts for amination among known hydrogenation catalysts (or their close analogues) can produce good results. Thus, the aim of this work was to study the applicability of a series of efficient Co-based hydrogenation catalysts and one new analogue for the amination of aromatic aldehydes, and to reveal the influence of the composition and the structure of the composites on their catalytic performance.

It was shown that pyrolysis of 3d metal complexes deposited on porous carriers could be a quick and simple method for the preparation of the efficient hydrogenation catalysts [24,29,31–33,41–44]. These catalysts were air stable and could be used in hydrogenation of organic compounds without special precautions, such as storage in an inert atmosphere. We recently reported the synthesis and properties of a series of composites, which efficiently catalyzed the hydrogenation of quinoline [45]. These composites were prepared by pyrolysis of $Co^{II}$ complexes with 1,10-phenantroline (Phen, the composite, is hereinafter referred to as Co-Phen/SiO$_2$), 1,2-diaminobenzene (DAB; Co-DAB/SiO$_2$-1 and Co-DAB/SiO$_2$-2) and melamine (Co-Mel/SiO$_2$). In this study, we tested the catalytic productivity of four Co-containing composites from the series, reported by us recently [45], in the amination of aromatic aldehydes. These four composites were selected because of their high efficiency in the hydrogenation of quinoline; i.e., among the studied series, use of these samples led to the highest yields of 1,2,3,4-tetrahydroquinoline [45]. We supposed that these systems would also be efficient in amination reactions. Abbreviations used for the definition of the composites are shown in Table 1.

**Table 1.** Abbreviations used for the definition of the composites together with Co content in these systems.

| Composite | Organic Precursor for Pyrolysis | $\omega$(Co) in the Composite, % by Weight | Ref. |
|---|---|---|---|
| Co-Im/SiO$_2$ | imidazole | 5.7 | this work |
| Co-Phen/SiO$_2$ [a] | 1,10-phenantroline | 3.7 | [45] |
| Co-DAB/SiO$_2$-1 [b] | 1,2-diaminobenzene | 8.0 | [45] |
| Co-DAB/SiO$_2$-2 [c] | 1,2-diaminobenzene | 5.3 | [45] |
| Co-Mel/SiO$_2$ [d] | melamine | 3.4 | [45] |

(a) sample corresponds to Co-Phen/SiO$_2$-1 reported in [45]; (b) sample corresponds to Co-DAB/SiO$_2$-1 reported in [45]; (c) sample corresponds to Co-DAB/SiO$_2$-2 reported in [45]; (d) Sample corresponds to Co-Mel/SiO$_2$-4 reported in [45].

Apart from these studies, an efficient Co-containing catalyst for the amination of aromatic aldehydes was prepared by pyrolysis of $Co^{II}$ acetate with 1-methyl-3-cyanomethyl-1H-imidazolium chloride on activated carbon [32]. It was supposed that the fragments of imidazole could retain their structure after pyrolysis [46], providing that the carboneous material, originating from imidazole, would be significantly different from these, prepared

by the pyrolysis of other organic precursors. Inspired by these findings, we synthesized a new composite by pyrolysis of $Co^{II}$ acetate and imidazole deposited on aerosil, which we report here as Co-Im/SiO$_2$. Thus, the series of Co-containing composites selected for this study was extended with a new composite Co-Im/SiO$_2$, prepared by the thermal decomposition of $Co^{II}$ complex with imidazole on the aerosil.

## 2. Results and Discussion

Thermal decomposition of $Co^{II}$ complexes deposited on various carriers led to the formation of the metallic cobalt nanoparticles together with the carboneous species [45]. In this study, three organic precursors were used, i.e., Phen, DAB and Mel. The first two organic precursors (Phen and DAB) contained more than 66% of C and less than 8% of N, and their thermal decomposition in argon quite expectedly led to the systems, containing N-doped carboneous material, together with nanoparticles of metallic cobalt. Carbon content in the resulting composites was in the range from 8.5 to 65% [45]. Use of melamine (29% of C, 67% of N) as a component in the starting "reaction mixture" additionally resulted in the formation of the catalytically active composites, which had a significantly lower content of C—from 0.2 to 4.5%—and a low content of N (<0.3%). Nevertheless, some representatives of all these three series possessed high catalytic productivity in the hydrogenation of quinoline, and the catalytic properties of the composite did not correlate with the C or N content [45].

Imidazole has an intermediate content of C and N (53 and 41%, respectively) compared to the previously studied Phen, DAB, from one side, and Mel, from the other side. The thermal decomposition of $Co(CH_3COO)_2$ in combination with imidazole, deposited on the aerosil, led to the formation of the black carbon-containing composite (details are provided in the Experimental section). Pyrolysis of $Co^{II}$ imidazolate led to the composite with a relatively high N content (1.54%) compared to other systems considered in this work (N content is in the range 0.13–0.3%), and the value of $\omega(N)/\omega(C)$ for Co-Im/SiO$_2$ (0.24; $\omega$ is weight%) was among the highest values observed for similar series [45].

It can be seen from the comparison of the compositions of the precursor mixtures ($Co^{II}$ acetate and the organic compound deposited on SiO$_2$ before thermal treatment) and the resulting composites, that the contents of C in the composites generally grew with an increase in the C content in the precursor samples (samples prior to pyrolysis, Figure 1; for the calculation of the precursors composition, the C content in $Co(CH_3COO)_2$ was not taken into account because thermal decomposition of this acetate led to the formation of volatile organic compounds, and solid residue mainly contains metallic cobalt and minor admixtures of carbon [47]). Such observations can provide evidence that the main part of the carbon from the starting organic compounds is transformed into carboneous particles in the composite (in contrast to gaseous reaction products). At the same time, a similar dependency was not found for nitrogen, and this observation can be explained by the fact that no component of the resulting composites contains N as the major component (N is a minor component of *N*-doped carboneous particles [48]).

The new composite Co-Im/SiO$_2$ was characterized by the powder XRD, TEM and Raman spectroscopy. The reflections at 2θ = 44.3°, 51.4°, 75.9° were found in an X-ray diffraction pattern of Co-Im/SiO$_2$ (Figure 2). These peaks can be assigned to Co (Fm $\bar{3}$ m) [49]. No peaks which could be assigned to other Co-containing compounds, such as oxides, were found.

Dark distinct nanoparticles of ca. 7–15 nm size were found on the TEM image (Figure 3). These nanoparticles could be assigned to metallic Co. The shape of the nanoparticles was close to spherical, and no aggregates were observed. The appearance of Co nanoparticles in Co-Im/SiO$_2$ is quite similar to the one found for the Co-DAB/SiO$_2$ and Co-Mel/SiO$_2$ series [45]. No distinct species which could be assigned to carboneous material could be found in the TEM image of Co-Im/SiO$_2$. This observation could be the sign of a finely dispersed carboneous component.

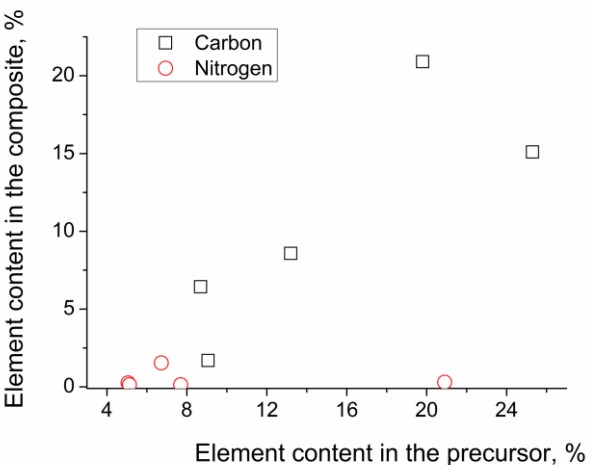

**Figure 1.** Content of C and N in the precursors before pyrolysis vs. content of these elements in the composites studied herein. Element content in the precursor was counted as weight of C in Im, Phen, DAB or Mel divided by summary weight of the anhydrous compounds multiplied by 100%, i.e., 100% $\times$ m(C$_{org}$)/{m(Co(CH$_3$CO$_2$)$_2$) + m(organic compounds) + m(SiO$_2$)}.

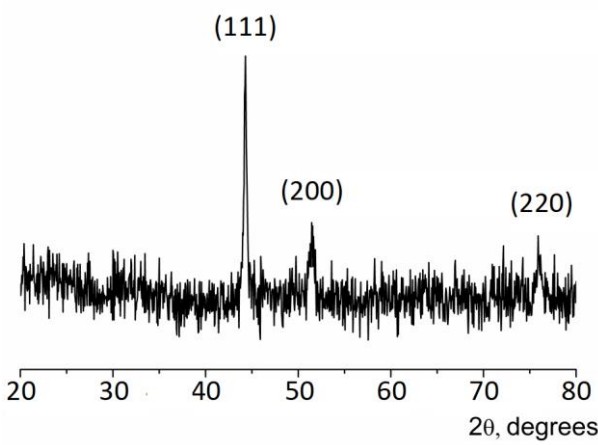

**Figure 2.** XRD pattern of Co-Im/SiO$_2$ composite.

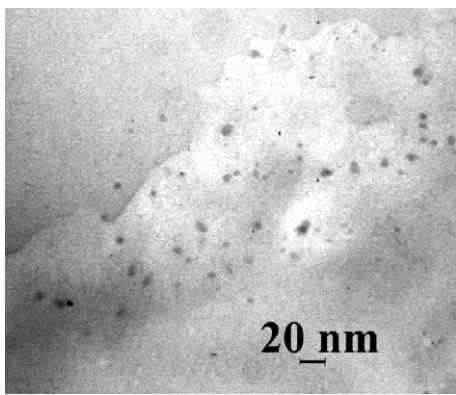

**Figure 3.** TEM image of the Co-Im/SiO$_2$ composite.

In the Raman spectrum of Co-Im/SiO$_2$, the bands at 1322 and 1587 cm$^{-1}$ (Figure 4) were detected. These bands could be assigned to characteristic D and G bands of disordered graphitic carbon, respectively [50]. A band at 890 cm$^{-1}$ can be assigned to Co nanoparticles [51]. The occurrence of the D band and the absence of the 2D band could be the signs of

non-ideal graphene [52,53]. The value of $I_D/I_G$ was 0.69, indicating significant disorder of the graphene layers [54]. It can be concluded that pyrolysis of $Co^{II}$ complex with imidazole led to the formation of the carboneous graphene-like disordered multilayer particles. The parameter $I_D/I_G$ for Co-Im/SiO$_2$ falls in the range of the values previously found for the composites, prepared by pyrolysis of other $Co^{II}$ complexes on the aerosil [45].

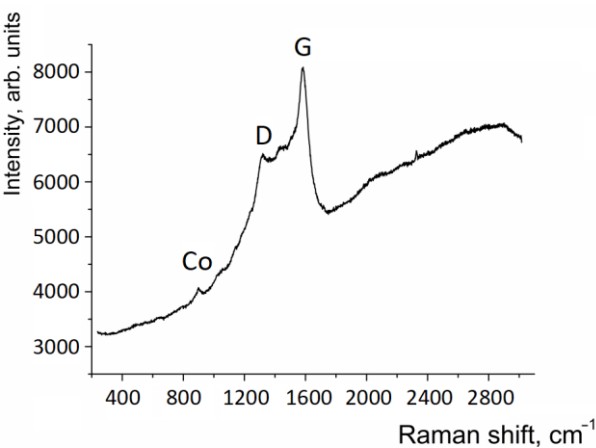

**Figure 4.** Raman spectrum of the Co-Im/SiO$_2$ nanocomposite.

In the preliminary experiments on catalytic amination, it was found that in the mixture containing *p*-methoxybenzaldehyde by *n*-butylamine at p(H$_2$) = 100 bar and T = 50 °C in methanol, with a catalyst loading 2 mol% for a 4 h reaction time, amination did not occur at all: in the cases of all composites, only the respective Schiff base was found (SC on Scheme 1). In the same reaction, at an increased pressure and temperature (p(H$_2$) = 150 bar and T = 150 °C) and an increased catalyst loading (5 mol% instead of 2%), quantitative formation of *N-n-butyl-N-p*-methoxybenzylamine (AM on Scheme 1) almost took place in the presence of all composites. It can be concluded that the composites had significant catalytic activity in the amination. In order to differentiate the composites and to reveal the one which was the most active in the amination, further experiments were carried out at p(H$_2$) = 100 bar and T = 100 °C in methanol, with a catalyst loading 3 mol% (Table 2).

**Scheme 1.** Reaction scheme of amination of *p*-methoxybenzaldehyde with amine R-NH$_2$, where R = *n*-butyl or benzyl. Reaction products: AM = *N*-substituted-*N*-(*p*-methoxybenzyl)amine, AL = *p*-methoxybenzyl alcohol, SC = Schiff base from *p*-methoxybenzaldehyde and amine.

**Table 2.** The yields of the products in reactions of amination of *p*-methoxybenzaldehyde to $H_2$ in presence of Co-containing composites together with the yields of 1,2,3,4-tetrahydroquinoline (THQ) in hydrogenation of quinoline in presence of these composites.

| | Product Yields for *n*-Butylamine (1.0 mmol) % [a] | | Product Yields for *n*-Butylamine (1.5 mmol) % [a] | Product Yields for Benzylamine (1 mmol) % [a] | Product Yields for Diisopropylamine, (1.5 mmol), % [a] | Yield of THQ [b], % |
|---|---|---|---|---|---|---|
| P, bar | 100 | 150 | 100 | 100 | 100 | 100 |
| T, °C | 50 | 150 | 100 | 100 | 100 | 100 |
| Co loading, mol.% | 2 | 5 | 3 | 3 | 3 | 3 |
| Co-Im/SiO$_2$ | SC 100 | AM 100 | AM 92 AL 8 | AM 92 AL 8 | AL 90 ALD 10 | 55 |
| Co-Phen/SiO$_2$-1 | SC 100 | AM 100 | AM 86 AL 2 SC 12 | AM 92 AL 6 SC 2 | AL 90 ALD 10 | 13 |
| Co-DAB/SiO$_2$-1 | SC 100 | AM 100 | AM 88 AL 12 | AM 92 AL 8 | AL 99 | 14 |
| Co-DAB/SiO$_2$-2 | SC 100 | AM 100 | AM 94 AL 6 | AM 96 AL 4 | AL 99 | 32 |
| Co-Mel/SiO$_2$-4 | SC 100 | AM 100 | AM 72 AL 19 SC 9 | AM 73 AL 27 | AL 99 | 16 |
| Co-Mel/SiO$_2$-4 [c] | | | AM 89% AL 3% SC 8% | AM 85% AL 4% SC 11% | AL 99 | |

(a) other reaction conditions: methanol, time 4 h, n(*p*-methoxybenzaldehyde) = 1 mmol. (b) other reaction conditions: methanol, time 4 h, n(quinoline) = 1 mmol. (c) 0.6 g of 3A molecular sieves were added per 1 mmol of the aldehyde.

In the case of all studied composites in reactions of *p*-methoxybenzaldehyde to *n*-butylamine in the conditions mentioned above, (p(H$_2$) = 100 bar and T = 100 °C in methanol, with a catalyst loading 3 mol%), a complete conversion of the starting aldehyde was observed. The highest yield of *N*-*n*-butyl-*N*-*p*-methoxybenzylamine was achieved in the presence of Co-Im/SiO$_2$ and Co-DAB/SiO$_2$-2 composites (92 and 94%, respectively), and a very similar performance (within experimental error) was found for Co-DAB/SiO$_2$-1 (target amine yield was 88%). In these cases, *p*-methoxybenzyl alcohol was the only by-product.

Unexpectedly, in the case of Co-Phen/SiO$_2$-1, the yield of *N*-*n*-butyl-*N*-*p*-methoxybenzylamine was relatively high (86%), but the formation of the target compound was accompanied by the presence of a significant quantity—12%—of the Schiff base (SC, Scheme 1). Furthermore, when the reaction was carried out in the presence of Co-Mel/SiO$_2$-4, the Schiff base was additionally found among the reaction products. It can be concluded that the catalytic activity of these two composites was not sufficient for hydrogenation of the Schiff base to the target amine. It is probable that the amination could pass through the formation of hemiaminal (not shown in Scheme 1), which could undergo direct hydrogenation to the target amine or dehydration to the Schiff base. The presence of the Schiff base together with the target amine can indicate that the hydrogenation of the former compound was slower compared to the hydrogenation of the respective hemiaminal.

The tendency of the composites' efficiency in the reaction of *p*-methoxybenzaldehyde and benzylamine was similar to the tendency of these composites' efficiency in the amination of the same aldehyde with *n*-butylamine. As in the case of *n*-butylamine, the conversion of the starting aldehyde on Co-Im/SiO$_2$, Co-DAB/SiO$_2$-1 and Co-DAB/SiO$_2$-2 was complete; *N*-benzyl-*N*-*p*-methoxybenzylamine formed in a high yield (92% or more) and *p*-methoxybenzyl alcohol was the only by-product. In the case of Co-Phen/SiO$_2$-1, 2% of the Schiff base was detected, though *N*-benzyl-*N*-*p*-methoxybenzylamine also formed in a 92% yield. The catalytic performance of Co-Mel/SiO$_2$-4 was the lowest; use of this catalyst led to the formation of the target amine in just a 73% yield; however, *p*-methoxybenzyl alcohol was the only by-product and no Schiff base was detected.

An attempt to perform amination of $p$-methoxybenzaldehyde with diisopropylamine was unsuccessful, and only $p$-methoxybenzyl alcohol—the product of aldehyde hydrogenation—was formed (Table 2). Complete conversion of the starting compound, $p$-methoxybenzaldehyde, was not achieved in the cases of Co-Im/SiO$_2$ and Co-Phen/SiO$_2$-1: in both cases, the reaction mixture contained ca. 10% of the starting aldehyde. Therefore, use of these composites in the cases when $n$-butylamine or benzylamine were present in the reaction mixtures led to complete conversion of the $p$-methoxybenzaldehyde, while the replacement of these amines by diisopropylamine created an obstacle even to the complete reduction of the starting aldehyde. It was shown that amination of the secondary amines with the aromatic aldehyde (benzaldehyde) readily took place (for example, on the Pd(OH)$_2$/g-C$_3$N$_4$ catalyst [19]), and that consequently there were no fundamental obstacles to aldehyde amination with diisopropylamine. The difference in the catalytic activity of the composites in the presence of various amines was apparently not caused by the difference in their basicity, because the pK$_b$ value for diisopropylamine (3.4) is between the pK$_b$ values for $n$-butylamine and benzylamine (3.2 and 4.7). However, the observation was that the result of the direct hydrogenation of the $p$-methoxybenzaldehyde to the $p$-methoxybenzyl alcohol depended on the presence of amine in the reaction mixture; catalyst activation or deactivation by specific amine could be one of the reasons. In addition, the amination of $p$-methoxybenzaldehyde with $n$-butylamine or benzylamine in the presence of Co-Mel/SiO$_2$-4 resulted in very similar yields of the target amines (72–73%), but a different distribution of the by-products: 19% of $p$-methoxybenzyl alcohol and 9% of the Schiff base in the first case, compared to 27% of $p$-methoxybenzyl alcohol in the second case. It should be concluded that amination of $p$-methoxybenzaldehyde by amines in the presence of the studied composites is a complex process, which probably consists of several competing reactions, and is sensitive to composition of the reaction mixtures; notably, that there was no correlation between the content of Co or C or N in the composites and the yields of the amination products. Taking into account that equal counting per Co loading of the composites in the reaction mixtures led to different yields of the amination products, the catalytic efficiency of these species depends not only on the quantity of Co.

In order to check if the amination could indeed pass through a step of SC formation, rather than alkylation of the amine by AL (formed directly from ALD, Scheme 1), three types of experiments were carried out. The addition of 3A molecular sieves for the trapping of water, which released upon formation of the SC, shifted the equilibrium in the reaction of ALD with the primary amine towards formation of the SC, and in this case higher yields of AM along with lower yields of AL were observed (the last two lines in Table 2). In a separate experiment, hydrogenation of the Schiff base from $p$-methoxybenzaldehyde and benzylamine was carried out in the presence of 3 mol% of Co-Im/SiO$_2$ at p(H$_2$) = 100 bar, T = 100 °C (methanol, 4 h). In this experiment, 95% of AM and 5% of AL formed, quite similarly to the result of the treatment of the $p$-methoxybenzaldehyde and benzylamine mixture (Table 2). However, a slightly higher yield of AM in the case of the hydrogenation of pure SC compared to the mixture of ALD and benzylamine, and this agreed with the result of the treatment of these starting compounds in the presence of 3A molecular sieves (Table 2): elimination of water from the reaction mixture favored SC formation; in addition, the quantity of AL occurring due to the hydrogenation of ALD was lower. Finally, the treatment of the mixture of AL with benzylamine in the presence of 3 mol% of Co-Im/SiO$_2$ at T = 100 °C (methanol, 4 h) did not result in the formation of AM. Thus, alkylation of the primary amine in these reaction conditions could be excluded, and AM formed as a result of the hydrogenation of the SC (notably, these experiments did not exclude the possible formation of the intermediate hemiaminal). All AL, detected in the reaction mixtures, formed due to the hydrogenation of ALD and it was a by-product, which could not participate in further transformations.

Catalytic performance of the studied composites in amination of $p$-methoxybenzaldehyde only in part correlated with their performance in the hydrogenation of quinoline (Table 2). The composites Co-Im/SiO$_2$ and Co-DAB/SiO$_2$-2, which were the most efficient in the

hydrogenation of quinoline, showed the best performance in the amination, while the composite Co-DAB/SiO$_2$-1, which showed good results in the amination, was not very efficient in the hydrogenation of quinoline. The difference between the performance of the composites in the amination of aldehydes and hydrogenation of quinoline could be explained by the different mechanisms of these reactions (in particular, but not limited to, by the difference in the sorption energy of the reactants on the active sites). In any case, Co-DAB/SiO$_2$ composites could be considered as a more universal catalyst for the reactions involving hydrogen, because their use led to better result in different reactions.

In order to reveal if the studied composites were suitable for amination of the compounds containing halogen in the aromatic cycle, the reaction of *p*-chlorobenzaldehyde with *n*-butylamine was studied (Scheme 2; reaction conditions are shown in the caption for Table 3). It was found that the conversion of the starting aldehyde was complete in all cases, but the reaction mixtures contained complex sets of different products; some of these are shown in Scheme 2. Nevertheless, in all cases the reaction mixtures contained more than 60% of *N*-butyl-*N*-*p*-chlorobenzylamine. Identified products included *bis*-(*p*-chlorobenzyl)ether, which could form directly from *p*-chlorobenzyl alcohol, or, more probably, by the hydrogenation of hemiacetal of *p*-chlorobenzyl alcohol and *p*-chlorobenzaldehyde. Unexpectedly, the content of *p*-chlorobenzyl alcohol—the product of direct hydrogenation of the starting compound—in the reaction mixtures was very low.

**Scheme 2.** Scheme of amination of *p*-methoxybenzaldehyde. R = *n*-butyl or benzyl. Reaction products: AM = *N*-substituted *N*-(*p*-methoxybenzyl)amine, AL = *p*-methoxybenzyl alcohol, SC = Schiff base from p-methoxybenzaldehyde and amine.

**Table 3.** The yields of the products in reactions of amination of *p*-chlorobenzaldehyde with *n*-butylamine by H$_2$ in presence of Co-containing composites. Reaction conditions: p(H$_2$) = 100 bar, T =100 °C, catalyst loading 3 mol% (per Co), methanol, time 4 h, 1.5 mmole of *n*-butylamine per 1 mmol of *p*-chlorobenzaldehyde.

| Catalyst | Product Yields % [a] |
|---|---|
| Co-Im/SiO$_2$ | Cl-AM 60; Cl-Al 0.5; ET 33; Cl-ET 1 |
| Co-Phen/SiO$_2$-1 | Cl-AM 68; Cl-AL 4; Cl-ET 17 |
| Co-DAB/SiO$_2$-1 | Cl-AM 89; Cl-AL 0.5; Cl-ET 7 |
| Co-DAB/SiO$_2$-2 | Cl-AM 74; Cl-AL 1; AM 2.5; Cl-ET 13 |
| Co-Mel/SiO$_2$-4 | Cl-AM 70; Cl-AL 7; Cl-ET 11 |

(a) sum of products is not 100% because some products were not identified.

The highest yields of the target amine, *N*-butyl-*N*-*p*-chlorobenzylamine, were found in the case of reactions performed in the presence of Co-DAB/SiO$_2$-1 and Co-DAB/SiO$_2$-2 catalysts. As in the case of the amination of *p*-methoxybezaldehyde, the composites prepared by the pyrolysis of Co$^{II}$ complex with 1,2-diaminobenzene were the most efficient based on the criteria of productivity and selectivity.

## 3. Materials and Methods

Hydrogen (99.99%) was purchased from Galogas Ltd. (Kyiv, Ukraine), aerosil A175 was purchased from UkrReaChim Ltd (Kyiv, Ukraine). All other reagents and materials were obtained from UkrOrgSintez Ltd. (Kyiv, Ukraine) and used as received. The solvents were purified according to standard procedures [55]. In separate experiments, methanol (solvent) was additionally driesd over 3A molecular sieves and amination was carried out in the presence of 3A molecular sieves for water trapping [55].

Elemental analyses were performed using a CarloErba 1106 instrument. Powder X-ray diffraction measurements were performed using a Bruker D8 Advance diffractometer with CuK$\alpha$ radiation ($\lambda$ = 1.54056 Å). TEM measurements were performed using a PEM-125K instrument (SELMI, Sumy, Ukraine) operating at 100 kV acceleration voltage. Samples were suspended in methanol upon ultrasonic irradiation for a minute, a drop of the suspension was put on a Cu grid (300 mesh), covered by a film of amorphous carbon, immediately after the end of the ultrasonic treatment. Raman spectra of the composites were measured using a Horiba Jobin-Yvon T64000 spectrometer (Co-Yr laser, $\lambda$ = 457 nm; the spectra were reordered with a one-stage monochromator and a CCD detector). Co content in the composites was determined by atomic adsorption using an iCE3500 spectrometer (Thermo Scientific, Waltham, MA, USA) with acetylene-air flame atomization.

Amination and hydrogenation (quinoline) experiments were carried out in a steel high-pressure reactor, equipped with a manometer, magnetic stirrer and temperature controller, similarly to the previously reported techniques [44,45].

$^1$H and $^{13}$C spectra of the reaction mixtures were measured on a Varian Unity Plus 400 spectrometer. An Agilent 1100 LCMSD SL instrument (chemical ionization (CI)) and an Agilent 5890 Series II 5972 GCMS instrument (electron impact ionization (EI)) were used for chromatomass analysis of the reaction mixtures. High-resolution mass spectra (HRMS) were recorded on an Agilent Infinity 1260 UHPLC system coupled to a 6224 Accurate Mass TOF LC/MS system.

### 3.1. Synthesis of Co-Im/SiO$_2$ Composite

Co(CH$_3$COO)$_2$·4H$_2$O (1.740 g, 6 mmol) and imidazole (0.600 g, 6.0 mmol) were stirred in ethanol (20 mL) for approximately 15 min at room temperature. Subsequently, the whole reaction mixture was heated to 60 °C and stirring was continued for an additional 1 h. Aerosil (1.824 g) was added, and the mixture was evaporated to a dryness state under continuous stirring at 60 °C for several hours. The violet sample was ground to a fine powder, which was transferred to a ceramic crucible. The crucible was placed in the oven, the oven was flushed with argon during 15 min and heated to 600 °C at a rate of 20 °C per minute, and held at 600 °C for 1 h under an argon atmosphere. The oven was then cooled down to a room temperature. Argon was constantly passed through the oven during the whole process. Elemental analysis of Co-Im/SiO$_2$: C = 6.43, H = 0.70, N = 1.54, $\omega$(Co)= 5.7% by weight.

### 3.2. General Procedure for the Amination

The composite (mass was adjusted according to Co content), methanol (10 mL), substituted benzaldehyde (1 mmol) and amine (1 or 1.5 mmol) were charged in a Teflon-lined stainless steel high-pressure reactor. The reactor was flushed three times with argon gas and then pressurized with H$_2$ gas. The reactor was then heated to a working temperature with continuous stirring of the reaction mixture at 600 rpm for 4 h. After this time, the reactor was cooled down to a room temperature and depressurized. The solid catalyst was then separated by the centrifugation; the organic liquid was evaporated on a rotary evaporator (this procedure led to the elimination of the excess of amine, if there was any). The final product was analyzed by the gas chromatography and NMR ($^1$H and $^{13}$C). Pure compounds were separated by chromatography on a silica gel (eluent—chloroform/methanol mixture 9:1 by volume).

### 3.3. NMR Data of the Products

*N-n*-butyl-*N-p*-methoxybenzylamine. $^1$H (400 MHz, CDCl$_3$), δ, ppm: 7.25 (dd, J = 8.8, 6.1 Hz, 2H), 6.88–6.82 (m, 2H), 3.79 (s, 3H), 3.73 (s, 2H), 2.63–2.60 (m, 2H), 1.53–1.47 (m, 2H), 1.38–1.30 (m, 2H), 0.92–0.89 (m, 3H). $^{13}$C NMR (126 MHz, CDCl$_3$), δ, ppm: 158.6, 132.3, 129.4, 113.7, 55.2, 53.3, 48.9, 32.1, 20.5, 13.9. The spectra are presented in the Supplementary Materials. GC/MS (ES-API): *m/z* = 194 [M + H]$^+$.

*N*-benzyl-*N-p*-methoxybenzylamine. $^1$H (400 MHz, CDCl$_3$), δ, ppm: 7.35–7.32 (m, 4H), 7.31–7.27 (m, 3H), 6.87 (d, J = 8.5 Hz, 2H), 3.85–3.79 (m, 4H), 3.77 (s, 3H). $^{13}$C NMR (126 MHz, CDCl$_3$), δ, ppm: 158.2, 139.7, 131.5, 128.9, 128.1, 127.9, 126.7, 126.5, 126.3, 113.6, 54.8, 52.5, 51.9. The spectra are presented in the Supplementary Materials. GC/MS (ES-API): *m/z* = 228 [M + H]$^+$.

*N*-butyl-*N-p*-chlorobenzylamine. $^1$H (400 MHz, CDCl$_3$), δ, ppm: 7.34–7.29 (m, 4H), 3.78 (s, 2H), 2.63 (t, J = 7.2Hz, 2H), 1.54–1.48 (m, 2H), 1.39–1.34 (m, 2H), 0.97–0.91 (m, 3H). $^{13}$C NMR (126 MHz, CDCl$_3$), δ, ppm: 138.6, 132.6, 129.5, 129.4, 53.2, 48.9, 32.0, 20.4, 13.9. The spectra are presented in the Supplementary Materials. GC/MS (ES-API): *m/z* = 198 [M + H]$^+$.

*N*-(4-methoxybenzyl)benzaldimine. $^1$H (400 MHz, CDCl$_3$), δ, ppm: 8.32 (s, 1H), 7.71–7.74 (m, 2H), 7.31–7.35 (m, 4H), 7.22–7.27 (m, 1H), 6.91–6.93 (m, 2H), 4.79 (s, 2H), 3.79 (s, 3H). $^{13}$C NMR (126 MHz, CDCl$_3$), δ, ppm: 161.7, 161.3, 139.6, 129.8, 129.4, 128.6, 128.4, 127.9, 127.9, 127.3, 127.1, 126.9, 113.9, 64.9, 55.3. The spectra are presented in the Supplementary Materials. GC/MS (ES-API): *m/z* = 225 [M]$^+$.

## 4. Conclusions

It was shown that the composites, containing nanoparticles of metallic Co and *N*-doped carboneous species on an aerosil carrier, were efficient catalysts for the amination of aromatic aldehydes in the presence of hydrogen and that these composites can be recommended for preparative amination of aromatic aldehydes with primary alkyl- and benzylamines. In particular, amination of *p*-methoxybenzaldehyde with *n*-butylamine could be performed with almost quantitative yields at 150 °C and a hydrogen pressure 150 bar at catalyst loading 5 mol.%. There is a high probability that the composites can be used for amination of a wide range of aldehydes and primary amines, which do not contain functional groups sensitive to reduction with hydrogen. Such catalysts can be considered as an alternative to the systems, based on platinum metals. The advantages of such catalysts include simple preparation (pyrolysis of Co$^{II}$ acetate and organic precursor, deposited on the aerosil) and a low price, together with the absence of toxic platinum metals in their composition. Due to the low price, Co-containing catalysts can be recommended for use without recycling.

Studies of amination of *p*-methoxybenzaldehyde at 100 °C and p(H$_2$) = 100 bar, 3 mol% catalyst loading allowed us to distinguish the most efficient catalysts. In these conditions, amination of *p*-methoxybenzaldehyde with *n*-butylamine and benzylamine resulted in the formation of the corresponding amines with the yields 72–96%, while amination of *p*-chlorobenzaldehyde with *n*-butylamine on these composites led to *N*-butyl-*N-p*-chlorobenzylamine in 60–89% yields. Within the studied series of composites, Co-DAB/SiO$_2$ composites could be considered as the most efficient catalysts for the different reactions involving hydrogen, i.e., hydrogenation and amination.

The results of this study can be useful for the development of new platinum-metal-free catalysts for the amination of aromatic aldehydes, as well as for other reactions involving hydrogen.

**Supplementary Materials:** The following supporting information can be downloaded at: https://www.mdpi.com/article/10.3390/chemistry5010022/s1, Figure S1. $^1$H NMR of *N-n*-butyl-*N-p*-methoxybenzylamine (400 MHz, CDCl$_3$); Figure S2. $^{13}$C NMR of *N-n*-butyl-*N-p*-methoxybenzylamine (126 MHz, CDCl$_3$); Figure S3. $^1$H NMR of *N*-benzyl-*N-p*-methoxybenzylamine (400 MHz, CDCl$_3$); Figure S4. $^{13}$C NMR of *N*-benzyl-*N-p*-methoxybenzylamine (126 MHz, CDCl$_3$); Figure S5. $^1$H

NMR of *N*-butyl-*N*-*p*-chlorobenzylamine (400 MHz, CDCl$_3$); Figure S6. $^{13}$C NMR of *N*-butyl-*N*-*p*-chlorobenzylamine (126 MHz, CDCl$_3$); Figure S7. $^1$H NMR of *N*-(4-methoxybenzyl)benzaldimine (400 MHz, CDCl$_3$); Figure S8. $^{13}$C NMR of *N*-(4-methoxybenzyl)benzaldimine (126 MHz, CDCl$_3$).

**Author Contributions:** Formal analysis, Y.L.L., M.O.I., O.O.P., D.M.V. and S.V.K.; Investigation, V.V.S., V.M.A., Y.L.L., M.O.I. and O.O.P.; Methodology, S.V.K.; Validation, S.V.R. and D.M.V.; Writing—original draft, Y.L.L. and S.V.K.; Writing—review and editing, O.O.P. and S.V.K. All authors have read and agreed to the published version of the manuscript.

**Funding:** This research received no external funding.

**Data Availability Statement:** The data presented in this study are available on request from the corresponding author.

**Acknowledgments:** The authors thank I.Ye. Kotenko for TEM measurements. The work was supported by the National Academy of Sciences of Ukraine and Enamine Ltd.

**Conflicts of Interest:** The authors declare no conflict of interest.

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
