# Peer review of "Catalytic Reductive Amination of Aromatic Aldehydes on Co-Containing Composites"

_chemistry, doi:10.3390/chemistry5010022_

Round 1

Reviewer 1 Report

In my opinion, it is a practical article but it needs to be considered the following comments before publishing.

The result in abstract part must be extended. The first line of the abstract is not reflecting in the manuscript.

Introduction: The manuscript does not illustrate great attention and activity in the field. The purpose of the study is not well expressed. At the end of the introduction, the necessity and novelty of the work must be expressed well. In the last paragraph, the work is supported by Reference # 37,

Please use the references for your methods.

Analytical methods: Why and how the said parameters were selected for this work. More specific details are needed to be added with use of latest reference. Use of some pictorial; diagram will be more elaborative for readers.

There is no discussion in the manuscript; the innovation of the study is not well expressed.

Add more results in the manuscript, 1H and 13C spectra, GCMS, HRMS, XPS etc.

Tables, all tables look like a part of literature, modify or provide new tables based on reactions

Figures: Improve the figures quality and leveling  

Future scope of this study can be added 

Author Response

Reviewer 1

COMMENT

The result in abstract part must be extended. The first line of the abstract is not reflecting in the manuscript.

REPLY

We changed the first sentence of the abstract in order to make more emphasis on the main points of the study. At the same time, we did not add new information to the abstract because of its limited length. In our opinion the abstract reflects all important achievements shown in the manuscript.

COMMENT

Introduction: The manuscript does not illustrate great attention and activity in the field. The purpose of the study is not well expressed. At the end of the introduction, the necessity and novelty of the work must be expressed well. In the last paragraph, the work is supported by Reference # 37,

REPLY

In order to highlight activity in this field, we added references to some recent studies devoted to amination. We also revised Introduction section and clearly formulated the aim of the study, and in our opinion the novelty of the work now is written more clearly.

COMMENT

Please use the references for your methods.

REPLY

We added some references to purification of solvents and for the method of amination. Other physical methods were applied by the common procedures, and do not require additional references.

COMMENT

Analytical methods: Why and how the said parameters were selected for this work. More specific details are needed to be added with use of latest reference. Use of some pictorial; diagram will be more elaborative for readers.

REPLY

We completely changed Table 1 with analytical data and added a pictorial instead of the part of the table (Fig. 1 in the revised version). Due to this change, we could make new conclusion about the relation between C content in the starting compounds and the composites. We also changed description of the reasons for selection of different starting compounds for preparation of the composites.

COMMENT

There is no discussion in the manuscript; the innovation of the study is not well expressed.

REPLY

We added discussion, and in particular we wrote the aim more clearly and added new information about experimental proofs of the reaction path. In our opinion innovation of the study is now more clear to the wide audience of readers.

COMMENT

Add more results in the manuscript, 1H and 13C spectra, GCMS, HRMS, XPS etc.

REPLY

We added NMR (1H and 13C) and MS data for the synthesized compounds in the main text, and we added figures of NMR spectra to the Supplementary materials. We did not perform XPS for a new composite, because previously we showed that the performance of these and similar composites in hydrogenation of quinoline did not correlate with the parameters of XPS spectra [Asaula, V.M.; Buryanov, V.V.; Solod, B.Y.; Tryus, D.M.; Pariiska, O.O.; Kotenko, I.E.; Volovenko, Y.M.; Volochnyuk, D.M.; Ryabukhin, S.V.; Kolotilov, S.V. Catalytic hydrogenation of substituted quinolines on Co-graphene composites Eur. J. Org. Chem. 2021, 2021, 6623-6632. https://doi.org/10.1002/ejoc.202101311]

COMMENT

Tables, all tables look like a part of literature, modify or provide new tables based on reactions

REPLY

We can agree with the Referee only regarding Table 1 – indeed, the majority of boxes in this table were filled by literature data. We completely changed it and added Fig. 1 instead of the main part of the table.

In Table 2, only the last column contained literature data. We replaced this column by a new one, based on our own data (because in reply to recommendation of Referee 2 we repeated experiments on quinoline hydrogenation in other conditions).

The remaining boxes of Table 2 and Table 3 contain only own data, based on the studied reactions.

COMMENT

Figures: Improve the figures quality and leveling  

REPLY

We changed figures (Fig. 2-4 in the revised version) and improved their quality, in particular we increased size of the letters.

COMMENT

Future scope of this study can be added 

REPLY

We extended Conclusions section and added our opinion on the future scope of the catalysts studied.

Reviewer 2 Report

In this work, the authors demonstrate the preparation of composites containing nanoparticles of metallic Co and N-doped carboneous species on aerosil carrier and their application as catalyst for reductive amination of aromatic aldehydes. In general, the article is organized and clear. However, some details should be corrected before publication. There are typos in the text, the references should be revised (for example: Hong, Z.; Ge, X.; Zhou, S. Underlying Mechanisms of Reductive Amination on Pd-Catalysts: The Unique Role of Hydroxyl 467 Group in Generating Sterically Hindered Amine. Int. J. Mol. Sci. 2022, 23, 7621 was cited twice) and it is missing important recent contributions about reductive amination involving transfer hydrogenation and hydrogenation.

Was the methanol treated (to eliminate water) before the reactions? The presence of remaining starting material and consequently, of alcohols might be the consequence of the hydrolysis an imine hydrolysis due to the water contained in Methanol. Besides that, the amination also produces water molecules which can also resulting in hydrolisis. In this case, I recommend the use of drying agent such as NaSO4, MgSO4, molecular sieves, etc.

I also suggest a better evaluation of the conditions and a table showing all the tested conditions. As far as I understood, few fixed conditions have been tested. Have you tried long reaction times and lower temperatures, catalysts and concentration? If yes, you should show them.

A better evaluation of the conditions can also help to explain the mechanism and the side reactions involved. It  can also turn the methodology more efficient and mild.

If the THQ were not tested in the same condition of the other substrates,               I suggest to remove these results, or present them using the same conditions.

I also suggest the NMR spectra inclusion of the pure synthesized compounds.

Author Response

Reviewer 2

COMMENT

In this work, the authors demonstrate the preparation of composites containing nanoparticles of metallic Co and N-doped carboneous species on aerosil carrier and their application as catalyst for reductive amination of aromatic aldehydes. In general, the article is organized and clear. However, some details should be corrected before publication. There are typos in the text, the references should be revised (for example: Hong, Z.; Ge, X.; Zhou, S. Underlying Mechanisms of Reductive Amination on Pd-Catalysts: The Unique Role of Hydroxyl 467 Group in Generating Sterically Hindered Amine. Int. J. Mol. Sci. 2022, 23, 7621 was cited twice) and it is missing important recent contributions about reductive amination involving transfer hydrogenation and hydrogenation.

REPLY

We removed duplicating reference and, as recommended by the Referee, added recent references on transfer hydrogenation for amination and recent examples of hydrogenation.

COMMENT

Was the methanol treated (to eliminate water) before the reactions? The presence of remaining starting material and consequently, of alcohols might be the consequence of the hydrolysis an imine hydrolysis due to the water contained in Methanol. Besides that, the amination also produces water molecules which can also resulting in hydrolisis. In this case, I recommend the use of drying agent such as NaSO4, MgSO4, molecular sieves, etc.

REPLY

We used anhydrous commercially available methanol and it was not specially treated before the experiments. We carried out additional experiments using 3A molecular sieves as a drying agent and found that addition of this component led to slight increase of the yield of the amination product. This finding can be explained by more favorable conditions for formation of the Schiff base, which was the intermediate in the amination process.

These results were added to the revised manuscript. 

COMMENT

I also suggest a better evaluation of the conditions and a table showing all the tested conditions. As far as I understood, few fixed conditions have been tested. Have you tried long reaction times and lower temperatures, catalysts and concentration? If yes, you should show them.

REPLY

The conditions for zero yield of the amination product and quantitative yield of the amination product were found, and this information was provided in the text of the manuscript. For convenience, in the revised version of the manuscript these data were added to Table 2. These conditions can be used for "guaranteed" quantitative amination, if required.

We specially identified the conditions in which the product yields were between 50 and 100 %, because such experiment allows to differentiate the catalysts (if the product yields are 100 % in all cases, all catalysts seem to have identical activity).

COMMENT

A better evaluation of the conditions can also help to explain the mechanism and the side reactions involved. It  can also turn the methodology more efficient and mild.

REPLY

In order to get some insight into the reaction mechanism, we carried out additional experiments:

- hydrogenation of the pure Schiff base, prepared in separate synthesis;

- hydrogenation of a mixture of aldehyde and primary amine at presence of water trap (3A molecular sieves), which favors to formation of the Schiff base;

- treatment of a mixture of the alcohol (the product of hydrogenation of the aldehyde) and primary amine, expecting that alkylation of the amine could occur.

It could be concluded that the amination passed through formation of the Schiff base as the intermediate, and alkylation of the primary amine by alcohol could be excluded.

These results were added to the revised manuscript.

COMMENT

If the THQ were not tested in the same condition of the other substrates,             I suggest to remove these results, or present them using the same conditions.

REPLY

We performed additional experiments and added a column with THQ yields in the same

COMMENT

I also suggest the NMR spectra inclusion of the pure synthesized compounds.

REPLY

NMR spectra of the pure synthesized products were included (brief description – in Experimental section; figures – in the Supplementary Information).

Round 2

Reviewer 1 Report

The authors have made sufficient modifications, and I suggest that this paper be accepted without further modification.

Reviewer 2 Report

The authors improved the manuscript significantly , therefore I do recommend the publication.